# Prefoldin Subunits and Its Associate Partners: Conservations and Specificities in Plants

**DOI:** 10.3390/plants13040556

**Published:** 2024-02-18

**Authors:** Yi Yang, Gang Zhang, Mengyu Su, Qingbiao Shi, Qingshuai Chen

**Affiliations:** 1Shandong Provincial Key Laboratory of Biophysics, Institute of Biophysics, Dezhou University, Dezhou 253023, China; gzhang@dzu.edu.cn (G.Z.); smy@dzu.edu.cn (M.S.); 2National Key Laboratory of Wheat Improvement, College of Life Sciences, Shandong Agricultural University, Tai’an 271018, China; sqb900918@163.com

**Keywords:** prefoldin (PFD), prefoldin-like (PFDL), chaperone, unconventional prefoldin RPB5 interactor (URI), plant development, cytoskeleton assembly, hormonal signaling, gene expression

## Abstract

Prefoldins (PFDs) are ubiquitous co-chaperone proteins that originated in archaea during evolution and are present in all eukaryotes, including yeast, mammals, and plants. Typically, prefoldin subunits form hexameric PFD complex (PFDc) that, together with class II chaperonins, mediate the folding of nascent proteins, such as actin and tubulin. In addition to functioning as a co-chaperone in cytoplasm, prefoldin subunits are also localized in the nucleus, which is essential for transcription and post-transcription regulation. However, the specific and critical roles of prefoldins in plants have not been well summarized. In this review, we present an overview of plant prefoldin and its related proteins, summarize the structure of prefoldin/prefoldin-like complex (PFD/PFDLc), and analyze the versatile landscape by prefoldin subunits, from cytoplasm to nucleus regulation. We also focus the specific role of prefoldin-mediated phytohormone response and global plant development. Finally, we overview the emerging prefoldin-like (PFDL) subunits in plants and the novel roles in related processes, and discuss the next direction in further studies.

## 1. Introduction

Molecular chaperones are a class of proteins that are ubiquitous from prokaryotes to eukaryotes. From the defined concept, chaperone proteins can bind noncovalently to nascent peptide chains or unfolded proteins, helping them to fold, assemble, transport, and degrade, but do not exist in their final conformation. Chaperonins are divided into two types: the group I chaperonins are present in the mitochondria, chloroplasts, and the bacterial cytoplasm; the group II components are found in the cytoplasm of archaea and eukaryotic cells [1,2]. One of these auxiliary co-chaperones is prefoldin (PFD), found in archaea and eukaryotes, whose best-known function is to assist in the folding of unnatural proteins (especially cytoskeletal proteins) by delivering them to group II chaperonins [3,4]. There are six prefoldin subunits (PFD1–PFD6) that make up the prefoldin complex (PFDc) in eukaryotes [5,6,7,8,9]. The discovery of prefoldins is more than 20 years old and was initially named as Gims (genes involved in microtubule biogenesis) in a screen for mutants that are synthetically lethal with a mutated yeast γ-tubulin strain [10]. It was later renamed as “prefoldin” because it presented homologs in archaea and eukaryotes and was characterized as a co-chaperone protein in the cytoplasm [4]. In contrast to eukaryotic prefoldin, the archaeal prefoldin was soon systematically identified in *Methanobacterium thermoautotrophicum* [11]. While prefoldin chaperones in yeast and mammals have been intensively studied in recent decades, systematic research on the prefoldin in plants began only as late as 2009, despite the early report of bioinformatic analyses [12,13]. In recent studies, prefoldin subunits or complexes have been identified in the regulation of gene expression, transcription, transcript splicing, and protein homeostasis and have further been linked to human disease and plant growth [3,7,9,14,15,16,17,18], indicating the versatile capacity of prefoldins in multiple organisms.

In addition to the typical PFDc, there is another prefoldin-like (PFDL) complex (PFDLc) present in all eukaryotes, but not in archaea. As that PFDc, human PFDLc is constituted by prefoldin subunits with additional variant prefoldin partners to form the heterohexamer [7,16,19,20]. Unlike typical PFDc, the PFDLc has the additional ability to participate in RNA polymerases assembly [21,22], to stabilize phosphatidylinositol-3 kinase-associated protein kinases [23], and to inhibit certain cytoplasmic enzymes [24], and may also provide a platform for the stabilization and maturation of intracellular multiprotein complexes [25]. A very recent study demonstrated the existence of PFDLc in *Arabidopsis thaliana* (Arabidopsis) [26], indicating the conserved process in species evolution.

In fact, prefoldins and prefoldin-like subunits or complexes play fundamental roles in cell life. As fixed-growth organisms, plants have attracted much attention to harmonize the external variable environment with intracellular events for better cell survival. Although the study of plant prefoldin lags behind by a decade compared with yeast and mammals, the conserved and specific roles of plant prefoldins have been gradually aroused in recent years, implying their great potential in the plant kingdom. In this review, we first present the structure of PFDc and PFDLc. We then analyze the dynamic landscapes of plant prefoldins related processes, from cytoplasm to nucleus, and from phytohormone to global development. Finally, we put an unconventional PFDL subunit URI (unconventional prefoldin RPB5 interactor) as a paradigm, to review its conservation and specificities.

## 2. Structure of the Prefoldin Complex and Prefoldin-Like Complex

Prefoldin subunits are small proteins consisting of two coiled-coil structural domains and four or two β-strands that form α-type and β-type hairpin structures, respectively (Figure 1A,B). Eukaryotes PFDc consists two different α-types (PFD3 and PFD5) and four different β-types (PFD1, PFD2, PFD4, and PFD6) to form a heterohexameric complex (Table 1) [8,15], whereas in archaea, each type is replaced by either two or four copies [5,11,27]. Although yeast PFDc is still customarily called as “Gim complex”, all eukaryotes’ PFDc forms a similar jellyfish-like conformation with a rigid β barrel backbone in the center, from which six flexible coiled tentacles extend (Figure 1C). These six tentacles extend in the same direction and able to bind a variety of nascent client proteins based on substrate affinities formed by hydrophobic cavities generated by intrinsic residues in the distal region [5,6]. Unlike other chaperones, PFDc does not hydrolyze ATP when it physical interacts with its client proteins, but undergoes a stranding period until it is transferred to the group II chaperonin, CCT (chaperonins containing T-complex polypeptide-1, also named as TRiC (T-complex polypeptide-1 ring complex)), for further folding [6]. In addition, PFDc also protects unnatural proteins when they are captured by CCT and in the releasing stage [5]. Finally, when PFDc comes into contact with CCT, immature and conformationally unstable proteins are naturally transferred to CCT due to its high affinity and PFDc is automatically released [28].

Plant PFDc also contains six different subunits as that in eukaryotes (Table 1) [29]. Interestingly, each subunit in Arabidopsis showing more similarity to the orthologs of the corresponding subunit in human than to other subunits in the same organisms [12]. Arabidopsis PFDc showed high similarity confirmation with that in human [29]. Furthermore, analysis of prefoldin gene families from 14 plant species showed that all subunit genes have their close orthologues [30]. Moreover, both plant and human subunits can complement the defective phenotypes by deletion of yeast prefoldins [10,13]. All these findings strength the conservative properties of plant prefoldins.

**Table 1 plants-13-00556-t001:** The canonical and prefoldin-like subunit in different organism and research advances in plants.

Type	Name in Arabidopsis	AGI * Number	Name in Human	Name in Yeast	Functions in Plant
prefoldin subunit					
α-type	PFD3	AT5G49510	PFD3; VBP1	Gim2	cytoskeleton [13]; chromatin remodeling [31]; salt stress [13]; PINs trafficking [32]; GA-dependent interaction [33]
	PFD5	AT5G23290	PFD5; MM1	Gim5	cytoskeleton [13]; chromatin remodeling [31]; salt stress [13]; PINs trafficking [32]; GA-dependent interaction [33]
β-type	PFD1	AT2G07340	PFD1	Gim6	cytoskeleton [29]; chromatin remodeling [31]
	PFD2	AT3G22480	PFD2	Gim4	cytoskeleton [29]; chromatin remodeling [31]
	PFD4	AT1G08780	PFD4	Gim3	cytoskeleton [18]; chromatin remodeling [31]; splicing [14]; cold acclimation [18]
	PFD6	AT1G29990	PFD6	Gim1	cytoskeleton [34]; chromatin remodeling [31]; PINs trafficking [35]
prefoldin-like subunit					
α-type	URI	AT1G03760	URI; RMP	Bud27	cytoskeleton; PINs trafficking [32]
	UXT	AT1G26660	UXT; STAP1	-	?
β-type	PDRG1	AT3G15351	PDRG1	-	?
	ASDURF	AT1G49245	ASDURF	-	?

* AGI: Arabidopsis Gene Identifier. The short line (-) represent it do not exist in this organism, and the question mark (?) represent the function is unknown.

Another special PFDLc exists in the eukaryote kingdom. Similarly, human PFDLc is constituted of α2β4 elements. Two α-types are replaced by two variants called UXT (ubiquitously expressed transcript) and URI (unconventional prefoldin RNA polymerase binding subunit 5 (RPB5) interactor), while four β-types are composed of the typical PFD2, PFD6, an additional PDRG1 (p53 and DNA damage-regulated gene 1), and another β-type PFDL called ASDURF (asparagine synthetase domain-containing 1 upstream reading frame) (Table 1; Figure 1D) [15,16,19,25,36]. Yet, it contains only the URI homologue Bud27, as the unique PFDL protein exist in yeast (Table 1) [37,38]. Although Bud27 may interact with PFD6 by yeast two-hybrid assay and affinity purification in *Saccharomyces cerevisiae* [21,39,40], there is currently no direct and valid evidence for the exact subunits that constitute the yeast PFDL complex (Figure 1D).

Very recent studies in Arabidopsis indicates the existence of PFDL subunits in plants [26,32]. Arabidopsis URI ortholog was firstly identified by forward genetic screening with altered auxin distribution [32]. Further analysis demonstrated that URI was the PFDL subunits and showed conserved roles in Arabidopsis [26,32]. Moreover, putative ortholog analysis identified *PDRG1*, *UXT*, and *ASDURF* genes (Table 1), and AP-MS assay demonstrated URI, PFD2, PFD6, PDRG1, and ASDURF as interactors of UXT [26]. The interactions between URI and PFD2/PFD6 were also verified by yeast-two-hybrid assay and co-immunoprecipitate assay [32]. All these findings confirm the presence of PFDLc in the plant kingdom, which is similar to that in humans (Figure 1D).

## 3. Dynamic Functions of Prefoldin Proteins in Plants

### 3.1. Proteins Folding by PFDc and Its Associate Partner CCT

#### 3.1.1. Prefoldin Subunits in Cytoskeleton Organization

Newly unfolded proteins easily aggregate to disturb the proteostasis [1,41,42]. Nascent actin and tubulin always spontaneously self-organize, so the PFDc protects them until they are correctly folded. Therefore, PFDc plays the fundamental role in cytoskeleton organization. Plant cytoskeleton networks are involved in massive developmental processes, including pollen tube growth, root hair elongation, trichome morphogenesis, response to light signals, salt, pathogen attacks, and other biotic or abiotic stresses [43,44,45,46,47]. In fact, defects in one of these subunits in Arabidopsis are sufficient to cause abnormal cytoskeleton (Table 1) and links to developmental defects, with the aggravated phenotype by multiple mutants [13,14,18,34]. All prefoldin subunit mutants except *pfd1* exhibited abnormal microtubule arrangement in root elongation zones or hypocotyl cells compared with the wild type; all prefoldin single or multiple mutants are more sensitive to the microtubule depolymerizing drug oryzalin; and all prefoldin single mutants are sufficient to decrease the α-tubulin level [13,18,29,34]. Notably, prefoldin subunits are involved in microtubule-associated processes only as part of the PFDc, since the sextuple mutant does not increase sensitivity to oryzalin or decrease the α-tubulin level [29], suggesting that the prefoldin subunits act as the PFDc that promote microtubule organization and mediate the associated phenotypes in plants. However, there is a lack of direct experimental evidence as to whether the prefoldin subunits or the PFDc are required for actin organization in plant cells.

The critical function of prefoldins for cytoskeleton-related phenotypes is conserved in various species. For example, reductions in functional prefoldins reduce levels of α-tubulin and microtubule growth rates, leading to abnormal distal tip cell migration in *Caenorhabditis elegans* [48]. Mutations in *Drosophila* allele result in abnormal spindle and centrosome deletions and disruption of neuroblast polarity due to tubulin instability [49,50]. These landscapes reveal the intrinsic and conserved role of prefoldins in animals and plants. Nevertheless, from another point of view, sessile plants’ adaptation to perpetual environmental challenges associated with cytoskeleton organization is a specific mechanism for the evolution of green plants. Further studies could also focus on the coordination between PFDc and environmental stress stimuli.

#### 3.1.2. CCT Chaperonin, the Protein Folding Partner with PFDc

CCT chaperonin is the typical PFDc-anchored target protein. CCT complex is a conserved multi-subunit complex consisting of two back-to-back rings, each containing eight related but distinct paralogous subunits (named CCT1 to CCT8), forming a complex of approximately 1000 kDa. Eukaryotic CCT complex mediates nascent client peptides folding by encapsulating them in its enclosed central cavity [51,52,53,54,55]. Actin and tubulin captured by PFDc absolutely require CCT for refolding so the cytoskeleton is much more dependent on this biomacromolecules. In human, dysfunction of the CCT chaperonin leads to severe disease symptoms such as neurological-related disorders and cancer, largely due to their dysregulation in protein proteostasis and aggregation [51,54,55,56,57]. With similar construction and function, the Arabidopsis CCT complex is also assembled from eight CCT subunits, except that CCT6 has two coding regions that are 96% identical [12,58]. Plant CCT deficient cells exhibited depletion of cortical microtubules, accompanied by reduced tubulin level due to protein degradation [58], which is similar to the disorganized arrangement of tubulin in the *PFD* gene depletion mutants [13,18,29,34]. Moreover, prefoldin subunits also involved in the pathogenesis of human neurodegenerative diseases and tumors [7,9,59], which reveals the tightly correlation with CCT. Thus, PFD-CCT could be the conserved module in controlling the protein folding for proteostasis in organisms, both in mammals and plants.

There is a specific role of CCT subunits in determining the maintenance of plant stem cells. A recent study found that intrinsic enrichment levels of different CCT subunits in root stem cells ensured protein folding activity and proteomic integrity, thereby conferring proteotoxic stress [60]. For example, *cct7-2* and *cct8-2* exhibit a shorter root and disturbed stem cell niche, especially abnormal cell division and QC identity [60] and defective cell-to-cell trafficking [61]. Interestingly, the co-chaperone PFDc is also highly expressed in stem cells [60], implying that the PFD-CCT module may co-regulate stem cell identity and root growth and development. Because one subunit of the CCT subunits is sufficient to cause lethality in Arabidopsis [61], new biotechnological approaches, such as RNAi or bioinformatics analysis, could be considered to dissect the mechanism of PFD-CCT module in plant plasticity development, and of course, this is a permanent topic need to be concerned.

In addition to the cytoskeleton, there is approximately 10% other cellular substrates bind to the CCT complex for folding [55,62]. These proteins are involved in many important cellular processes, including cell cycle, cell division, DNA maintenance, metabolism, RNA processing, etc. [55]. In recent years, the catalytic subunit of protein phosphatase 4 (PP4c) has emerged as a candidate novel substrate for plant CCT; another protein, Tap46, appears to stabilize PP4c and prevent its degradation when bound to the CCT complex [58]. This finding implying the possibility linking CCT to protein phosphatase status, which may be associated with plant metabolism and growth.

### 3.2. Prefoldins in Nucleus

The previous section focuses on the function of prefoldin as a co-chaperone with CCT chaperonin for protein folding, especially for tubulin organization; clearly, this process occurs in the cytoplasm. It should be emphasized that plant prefoldins are also localized into nucleus [14,18,33]. Although prefoldins cannot act directly as transcription factors due to their lack of ability to bind DNA, they are also required in animals and plants to mediate nuclear gene expression in various aspects of the regulatory hierarchy. Here, we provide examples to fully understand their functions in various aspects.

#### 3.2.1. Transcription Regulation

Lots of studies put forward the critical role of prefoldins involved in transcript gene expression. Excess PFD1 in human lung tumors triggers its aggregation in the nucleus and binds to the transcription start site of the *Cyclin A* gene, repressing its expression [63]. The yeast prefoldin subunits bind to chromatin and affect RNA polymerase II activity, which regulate the transcription elongation [64]. Other in-depth studies have found that human PFD5 can act as a bridge protein to recruit other transcription co-repressors and functions as a tumor suppressor [65,66,67]. In addition, PFD6 directly binds to the FOXO (forkhead box O) transcription factor and enhances its transcriptional activity, thereby modulating the longevity response of *Caenorhabditis elegans* [68]. Therefore, nuclear prefoldins appears to regulate gene expression either by directly affecting the activity of core transcription factors or indirectly by recruiting other factors to regulate downstream cellular events.

Transcriptomic studies and ChIP-seq experiments in the model plant Arabidopsis demonstrate that prefoldins affect multiple groups of gene profiles [14,29,31]. This mechanism is much dependent on the requirement of prefoldins in chromatin remodeling (Figure 2A). SWR1c (SWI2/SNF2-related 1 chromatin remodeling complex) is the highly conserved complex to promote the exchange of H2A/H2B dimers with H2A.Z/H2B, which affects nucleosome stability and chromatin structure [69,70,71,72]. Notably, all prefoldin subunits interact with at least one of the components of SWR1c, indicating PFDc associate with chromatin remodeling to mediate gene expression (Table 1) [31]. More interestingly, the contribution of prefoldins to H2A.Z deposition corresponds to environment responsive genes [31], further strengthening the notion that prefoldins in the environmental adaptation. In next studies, it is necessary to dissect how prefoldins regulate SWR1c activity, and more importantly, to occupy the specific location in a set of gene loci responding to the environmental challenge.

#### 3.2.2. For Pre-mRNA Splicing

Pre-mRNA splicing is a fundamental process to expand proteome of eukaryotic organisms in post-transcriptional level [73,74]. In Arabidopsis, the key component of LSM2-8 (Sm-like 2–8) spliceosome complex is encoded by *LSM* genes, and *PFDs* are tightly co-expressed with *LSMs* [14]. Besides, all six prefoldins interact with LSM8, a specialized subunit required for LSM2-8 complex formation; this interaction maintained the adequate level of LSM8, and therefore, mediated the activity of the LSM2-8 spliceosome in response to U6 snRNA levels and pre-mRNA splicing patterns (Figure 2B) [14]. Interestingly, PFD4 acts as a co-chaperone associated with HSP90 that regulate LSM8 activity and mediate their interaction (Table 1) [14]. Although it is unclear whether PFD4 interacts directly with HSP90 or requires another co-chaperone, there is clearly an essential role for prefoldins involved in pre-mRNA splicing events in plant, which may open the door to a role for molecular chaperones in pre-mRNA splicing regulation. In line with this, human prefoldins act locally in the genome to regulate co-transcriptional pre-mRNA splicing [17], implying that prefoldins mediate the transcriptional splicing across different organisms.

#### 3.2.3. Protein Stability Regulation

Typical prefoldins are involved in gene expression not only by regulating the transcription or post-transcription processes, but also by mediating proteins stability. For instance, human PFD5 promotes c-Myc degradation by recruiting the proteasomes and a novel ubiquitin E3 ligase [75]. Another typical example is that prefoldins mediate the expression of viral genes by facilitating the interaction of HIV integrase with the VHL (von Hippel–Lindau) ubiquitin ligase, triggering its polyubiquitination and degradation [76]. In the case of plants, we enumerate a vital example of prefoldin-mediated low temperature acclimatization. HY5 (ELONGATED HYPOCOTYL 5), a member of bZIP (basic leucine zipper) transcription factors that integrates various processes including light, hormones, nutrients, and stress conditions [77,78,79,80,81], could be regulated by prefoldins at the post-translational level [18]. When plants perceive the cold stimulation, nuclear HY5 promotes the expression of cold-inducible transcripts, including anthocyanin biosynthesis genes, whereas prefoldins accumulate into nucleus and the interaction of PFD4-HY5 triggers HY5 ubiquitination and degradation, which in turn reduce its protein level and ensures the proper products of anthocyanins under cold stress (Table 1) (Figure 2C) [18]. Overall, this feedback loop ensures the balance transcripts in response to cold conditions. Remarkably, this nucleocytoplasmic shuttling of prefoldins is DELLA-dependent, a transcriptional regulator found in plants but not in other animals [82]. Therefore, although prefoldins link gene expression by modifying proteins stability in both humans and plants, distinct regulators exert the specific mechanism in different organism.

### 3.3. Special Correlation of Prefoldins to Plant Hormones

Plant hormones have been extensively studied in recent decades. These low-concentration molecules produced by plant metabolism are widely involved in plant growth, development, and stress response to the environment. Among them, the gibberellic acid (GA) and auxin are two important regulators of plant growth related to prefoldins. Here, we focus on the mechanism of prefoldin to GA signaling pathway and auxin transporters trafficking in plant cells.

#### 3.3.1. GA Signaling

The two most critical components of GA signaling include the GA receptor GID1 (gibberellin insensitive dwarf 1) and DELLA growth inhibitors. The current model of GA signaling suggests that GID1 senses GA signal and binds to DELLA to form the GA-GID1-DELLA trimeric complex, which subsequently triggers the polyubiquitinylation of DELLA and its degradation via the 26S proteasome pathway [82,83,84]. Therefore, when the GA level is high, DELLA is degraded and switches on GA signaling to promote plant growth; conversely, DELLA inhibits plant growth. Furthermore, DELLA integrates other hormonal signals and environmental cues to regulate plant growth [82,84,85]. The current study reveals a novel mechanism by which PFDc integrates GA-DELLA signaling and microtubules-mediated cell wall expansion. According to the yeast two-hybrid screen, PFD3 and PFD5 were captured as DELLA-interaction proteins (Table 1); this interaction triggers the accumulation of prefoldins in the nucleus and regulates their equilibrium level between the cytoplasm and the nucleus [33]. Notably, the localization of the entire PFDc rather than individual subunits into the nucleus depends on the presence of DELLA, although only PFD3/PFD5 interact with DELLA [33]. Thus, in the presence of GA, PFDc loses its ability to interact with DELLA because of its degradation and stays in the cytoplasm and functions as the cytoskeleton folding co-chaperone, whereas in the absence of GA, PFDc shuttles into the nucleus and severely compromises microtubules heterodimer. Therefore, PFDc is linked to GA signaling and regulates the balance of cell wall expansion through DELLA-dependent subcellular localization. Furthermore, the interaction between prefoldins and DELLA appears to be conserved, as PdPFD2.2 interacts with a putative DELLA in *Populus deltoides*, and the localization of PdPFD2.2 is regulated by this interaction [86]. Moreover, there are five GA responsive elements upstream of the *PdPFD2.2* gene locus, suggesting the possibility in responsive to GA [86]. Interestingly, overexpression *PdPFD2.2* increased biofuel conversion, which provides potential for bioenergy utilizing in the crop *Populus* [86,87].

#### 3.3.2. Auxin Signaling and Transport

A number of clues suggest a correlation between prefoldins and auxin signaling. Firstly, functional network analysis of maize prefoldins mapped a number of candidate interactors, including ABP1 (auxin-binding protein 1); in addition, 13 prefoldin genes in maize showed different responses to IAA treatment, with 3 genes up-regulated, 2 unchanged, and others down-regulated; furthermore, there are multiple ARF (auxin response factor) binding sites, auxin-inducible elements, and other hormone- or stress-inducible elements upstream in prefoldin genes locus [30]. All these findings implying that prefoldins may associate with auxin signaling. However, the delicate molecular mechanisms between prefoldin and auxin signaling pathways are still puzzling, so revealing their relationship is important and interesting for further work.

A unique property of auxin is the targeted cell-to-cell transport to establish the auxin gradients and maxima. Auxin transporters, including PINs (PIN-FORMED), exert the critical role in this process [88,89,90]. In particular, PINs turn out to circulate continuously between the PM (plasma membrane) and endosomes, which has been termed as PINs trafficking; this trafficking mechanism include the recycling pathway and the degradation pathway. In the former, PM-localized PINs are continuously endocytosed, and subsequently, returned to the PM with the secretion of the GNOM protein [88,90,91,92]; in the latter, PINs are degraded via endosome-mediated vacuolar targeting in a manner that is dependent on ubiquitin modification [92,93]. An effective way to monitor the cycling of PINs is BFA (brefeldin A) treatment, a fungal toxin that triggers the aggregation of PINs to form the so-called “BFA compartments” [91,94]. Interestingly, prefoldin mutants reduce the incidence of PIN2 in the PM [35]; this event may be related to defects in the transport of PIN2 from the PM to the endosome or back to the PM, as BFA treatment reveal extensive BFA compartment in *pfd3* or *pfd5* mutants [32]. Therefore, prefoldin subunits can link the distribution of auxin by regulating polarity localization of auxin transporters (e.g., PINs), at least in their mode of transport (Table 1).

#### 3.3.3. Are Prefoldins Required for the Coordination between GA and Auxin?

Multiple studies have revealed the tightly relationship between GA and auxin. For example, auxin is required to promote GA signaling response [95]. In turn, GA modulates the transport of PINs to regulate the auxin gradient, as evidenced by the fact that high GA levels required for PINs to circulate back to the PM and inhibit their vacuolar targeting, whereas low GA levels promote their vacuolar transport [96,97]. Besides, some ARFs may be associated with DELLA, and thus, massive downstream components could be regulated [98,99]. Therefore, one concerned question aims that the coordination between prefoldins and phytohormone crosstalk. Notably, prefoldins do action in GA-facilitated PINs transport [35], indicating that prefoldins are required in the crosstalk between auxin and GA pathways and further regulate hormone-mediated plant development. Further studies could focus on the functions of prefoldins in other hormones, which is a special manner to distinguish those animals.

### 3.4. Global Regulation for Plant Growth in a PFDc-Dependent or -Independent Manner

To dissect the different processes regulated by PFDc or individual subunits under conditions of plant development or environmental stress, a recent study generated a sextuple mutant by crossing that were defective in the activity for each subunit [29]. The antagonistic regulation between GA and abscisic acid (ABA) signal is thought to be a major determinant in the control of seed germination [100,101]. An earlier study indicates that PFD4 maybe one candidate ABI3 (abscisic acid insensitive 3)-interacting partners [102], but it lacks validation later. Recent transcriptomic data also enrich germination-related terms in the *6×pfd* mutant, and all prefoldin genes are transcriptionally active in imbibed seeds [29]. Therefore, prefoldin subunits may be involved in seed germination through an ABA-dependent signaling pathway. However, this process may be dependent on the entire PFDc rather than individual subunits, as all *pfd* single mutants exhibited a similar delay in germination compared with the *6×pfd* mutant [29].

The growth of hypocotyl and root are also much dependent on the canonical complex, as the *6×pfd* mutant did not show any exacerbated phenotype. Consistent with this, the flowering time is regulated in a PFDc-dependent manner. However, the growth of another organ, the rosette leaf, may be regulated in at least two ways because of the smaller size of the rosette leaf in the *6×pfd* compared with the single *pfd* mutant [29]. Therefore, there are at least two ways to regulate plant growth in different scenarios.

Prefoldins are also involved in abiotic stress adaptation. Earlier studies showed that the yeast *pfd* mutants are hypersensitivities to salt stress [10,13,103]. With parallel evolution, all prefoldin subunits in Arabidopsis are required for root growth under salt stress (100 mM NaCl). Interestingly, this response may be Na^+^-specific because neither ionic nor osmotic stress requires each subunit [29]. It is possible that the PFDc is involved in salt stress response in a manner associated with cytoskeletal reorientation, as salt stress induces depolymerization and reorganization of cortical microtubules [104,105,106,107]. Another abiotic stress response, cold acclimation, is also dependent on the entire PFDc, as any stacking effect is lacking in the *6×pfd* mutant [29].

The individual prefoldin subunit is much exert to regulate gene expression. Transcriptomic data showed that more differentially expressed genes (DEGs) were found in *6×pfd* compared with *pfd4* single mutant (a total of 1186 DEGs in *6×pfd*, while only 198 DEGs in *pfd4*) [14,29], highlighting the relevance of the individual subunit of the prefoldin, or the alternative complex, rather than the entire hexamer, in mediating gene expression. Interestingly, the DEGs in the *6×pfd* can be divided into three gene clusters and each of which is associated with a number of key transcription factors [29]. Thus, single prefoldin subunit or alternative recombinants are required to regulate gene expression for plant development.

## 4. The Unconventional Prefoldin RPB5 Interactor, a Representative PFDL Subunit

In addition to the classic prefoldin subunits, there are also PFDL members exist in eukaryotes kingdom. Human and plant PFDL subunits include URI, UXT, PDRG1, and ASDURF [26,36], while Bud27 (URI homologue) is the exclusive PFDL in yeast [38]. Here, we put plant URI as the paradigm and summarize the conservations and specificities compared with that in animals.

The URI and its orthologues across species contain three conserved regions known as the PFD domain, the RPB5-binding domain, and the URI box (Figure 1B) [32,37,38,108]. The PFD domain is highly conserved and thought to be required for interaction with PFDL modules, therefore forming the PFDL complex [19]. The mutation in the PFD domain of Arabidopsis URI eliminate its close interaction with PFD6 also strength this notion [32]. In fact, this interaction is associated with the cytoskeleton arrangement and auxin transport, at least by PINs trafficking (Table 1) [32].

Downstream of the PFD domain is a large extend fragment and so called as the intrinsically disordered region [26]. The disordered structure is important for various effector functions, because the flexibility region provide a platform to interact with other partners or undergo protein modifications [109,110]. In *Chlamydomonas reinhardtii*, some intrinsically disordered proteins are related to unfolded protein binding or chaperone function [111], indicating the tightly association between molecular chaperones and disordered regions. Nevertheless, this unique region endowed URI more functions beyond as a co-chaperone. For example, the RPB5-binding domain in this region is essential for RNA polymerase-related events. Both the human URI and its yeast ortholog Bud27 bind and regulate the stability of RPB5, a common subunit of three eukaryotic RNA polymerases, affecting RNA polymerase assembly or activity [19,21,22,112]. Yeast Bud27 also regulates RNA polymerase-dependent transcription and ribosome biogenesis processes [113,114]. In Arabidopsis, URI has an extensive interactome than UXT, another PFDL subunit that lack the extended region downstream in PFD domain, and more importantly, most of the URI-associate partners are related to RNA-binding or metabolic processing [26]. In conclusion, the RPB5-binding domain confers a variety of functions not found in typical prefoldins. It is interesting to explore how plant URI facilitates RNA polymerase related processes, such as ribosomal biogenesis, transcription, or translation.

As for the last “URI box”, which encodes a short α-helix peptide at the C-terminus (Figure 1B), little is known about the exact function. A few animal studies imply that it may be required for transcription or translation [39,115]. However, the URI box in plants has not been accurately annotated.

In human, URI also functions as a signaling protein by phosphorylation transformation. For example, URI phosphorylation prevents its interaction with PP1γ (protein phosphatase 1γ) [24], and the released URI could regulate O-linked N-acetylglucosamine transferase activity [116], both of which regulate cell survival. Particularly, there are 13 phosphorylated Ser/Thr residues in Arabidopsis URI, and these phosphorylation residues are required to maintain its stability [26]. Interestingly, all of them are in the disordered region, which is in line with two phosphorylated sites in human [19,26]. Further work could focus on the mechanism of plant URI phosphorylation and the linkage of this phosphorylation transformation to cell life.

## 5. Conclusions and Future Directions

Generic prefoldin chaperones are not only involved in cytoskeleton folding together with their associated partners (e.g., CCT), but are also required for protein stability and gene expression regulation at the transcription or post-transcription level. The role of plant prefoldins have some specificities compared to the progress in animals. In this review, we summarize the conserved roles of prefoldin in the structure of the prefoldin complex, the associated protein CCT, and cytoskeleton folding processes. In addition, we focus on the specific roles of prefoldin in plant development and give some paradigms for a comprehensive understanding. These roles involve coordination with hormonal signals, such as GA and auxin, PFDc-dependent or independent responses to abiotic stress, and regulation of environment adaptation by nuclear prefoldins. In addition to the classical prefoldin subunits, other PFDL members have gained attention in recent years. The URI prefoldin-like subunit appears to be required for a wider range of cellular processes according to the extend region. Certain aspects of prefoldin-related modes of action require further research.

For the prefoldin subunits, some of the following issues can be addressed: (i) what transcription factors can be recruited and how do they coordinate gene expression with prefoldins; (ii) since prefoldin is required for the GA and auxin pathways, how does it mediate upstream environmental signals and regulate intracellular biosynthesis or transport; (iii) why are there PFDc-dependent and -independent ways during plant growth or abiotic stresses, and what is the intrinsic mechanism of substituting subunits during these processes; and (iv) the molecular mechanism by which prefoldins work together with their co-chaperone proteins to promote protein stability remains undermined.

For prefoldin-like subunits, some questions remain to be answered: (i) how do the URI contribute to RNA polymerase-related processes, such as ribosome synthesis, transcription initiation/elongation, or translation in plants, and what are the specific roles of the URI in plants; (ii) does URI manipulate those cellular events by its individual subunit or PFDL complex; and (iii) what kind of proteins could be recruited and regulate URI phosphorylation and how do they work. All these questions need to be seriously considered.

## Figures and Tables

**Figure 1 plants-13-00556-f001:**
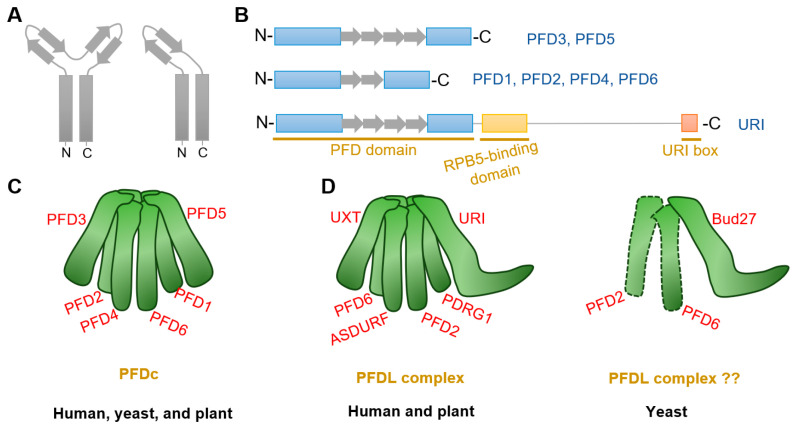
Structure of PFDc and PFDLc. (**A**,**B**) Structure of α-type (including PFD3 and PFD5) and β-type (including PFD1, PFD2, PFD4, and PFD6) prefoldins. Rectangles and arrows represent coiled-coil domain and β strands, respectively. α-type prefoldins contain four β strands to form two hairpins, whereas β-type prefoldins construct only one hairpin. In addition to the typical prefoldin subunits, another PFDL named URI constitutes with external RPB5-binding domain and a URI box. (**C**) Model of canonical PFDc in human (*Homo sapiens*), yeast (*Saccharomyces cerevisiae*), and plant (*Arabidopsis thaliana*). Note that all species showed similar PFDc conformation, although yeast is customarily named “Gim” (see Table 1). (**D**) Model of PFDLc in human, yeast, and plant. Note that human and plant PFDLc display similar confirmation, while there is no direct evidence for the existence of the yeast PFDL complex.

**Figure 2 plants-13-00556-f002:**
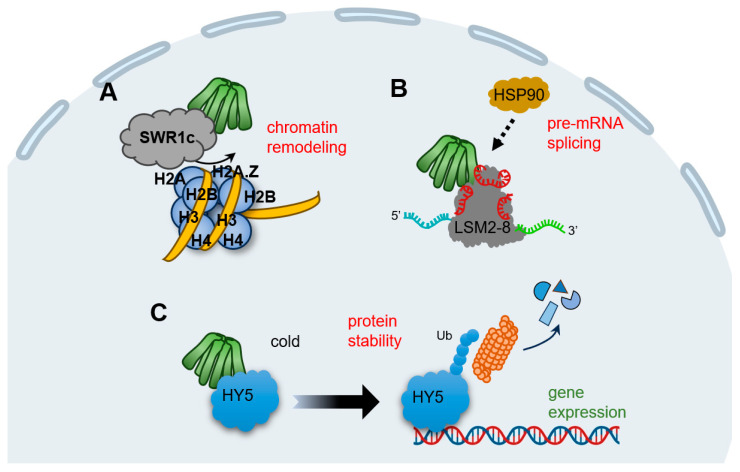
Schematic representation of the known plant PFDc in nucleus. (**A**) Prefoldins associate with SWR1c to favor H2A/H2B into H2A.Z/H2B to affect chromatin structure. (**B**) Prefoldins interact and mediate the activity of LSM2-8 spliceosome and then direct the pre-mRNA splicing. PFD4 may be a subunit to recruit HSP90. (**C**) In cold stimulation, HY5 promotes the expression of cold-inducible transcripts, while PFD4 interact with HY5, triggering its ubiquitination and degradation, to attenuate this process and ensure the proper products of downstream transcripts.

## Data Availability

Data from this study are available in this article.

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
