# Peer review of "Prefoldin Subunits and Its Associate Partners: Conservations and Specificities in Plants"

_plants, 2024, doi:10.3390/plants13040556_

Round 1
Reviewer 1 Report
Comments and Suggestions for Authors
The manuscript is about the presence, function, and conservation of prefoldin-like proteins in plants where the authors highlighted the roles of prefoldins in different processes such as protein folding, cytoskeleton organization, transcription regulation, and plant growth and development. The manuscript has an attractive topic and it can attract many readers in plant biology-related fields. This manuscript can be suitable for publication by considering some minor issues listed below;
1- Please add some data about the effects of prefoldin on microalgae as a group of plants.
2-Please explain how prefoldin contributes to plant development, cytoskeleton assembly, hormonal signaling, and gene expression.
Comments on the Quality of English Language
English style is good enough.
Author Response
Comments 1 & 2: The manuscript is about the presence, function, and conservation of prefoldin-like proteins in plants where the authors highlighted the roles of prefoldins in different processes such as protein folding, cytoskeleton organization, transcription regulation, and plant growth and development. The manuscript has an attractive topic and it can attract many readers in plant biology-related fields. This manuscript can be suitable for publication by considering some minor issues listed below:
1- Please add some data about the effects of prefoldin on microalgae as a group of plants.
2- Please explain how prefoldin contributes to plant development, cytoskeleton assembly, hormonal signaling, and gene expression.
Response 1: Thank you for pointing this out. This is indeed an important emerging filed in microalgae. In fact, up to now, there is no published research about prefoldin chaperones in microorganisms. Despite this, we have searched articles about disordered proteins linked chaperones (doi: 10.1038/s41598-018-24772-7) and HSP chaperones linked abiotic stress (doi: 10.1111/tpj.12816; doi: 10.1016/j.ecoenv.2013.11.015; doi: 10.1016/j.jplph.2016.07.012) in the green alga Chlamydomonas reinhardtii. For the former, we have added the data in the paragraph corresponding to “intrinsically disordered proteins” (lines 410-413 in the revised manuscript). For the latter, because of this review talk about prefoldin chaperones rather than HSPs, we considered that it is not very suitable to combine them into our manuscript. It is necessary to discuss this topic in the next article to systemic summarize HSP chaperones.
Response 2: Many thanks for this reviewer’s assessment.
For prefoldins (PFD) to global plant development, it relays at least two manners: PFD complex-dependent or single/alternative subunits dependent. The seed germination, hypocotyl and root growth, and flowering are much dependent on the canonical complex. However, the growth of another organ, the rosette leaf, may be not absolutely required the whole complex. These scenarios we have described in the section of “3.4” in the re-submitted manuscript. However, the in-depth molecular mechanism to distinguish the entire PFD complex (PFDc) and single/alternative subunits remains elusive. Of course, this aspect needs to be concerned in the next step and we point it in the section of “future directions” (see highlight lines 457-459).
For PFD to cytoskeleton assembly, it absolutely relays CCT chaperonins. Newborn cytoskeleton peptides captured by PFDc are transferred to the group II chaperonin, CCT, for further folding. The mechanism of the interaction between PFDc and cytoskeleton is much dependent on the substrate affinities (see highlight lines 79-80). Then, when PFDc comes into contact with CCT, immature and conformationally unstable cytoskeletons are naturally transferred to CCT due to its high affinity and PFDc is automatically released (see highlight lines 81-88). As for CCT folding, it needs to encapsulate the unstable proteins in the central cavity of CCT complex body (see highlight line 169-170).
For hormonal signaling, we mainly enumerated GA and auxin. The current model of GA signaling puts that GID1 senses GA signal and binds to DELLA to form the GA-GID1-DELLA trimeric complex, which subsequently triggers the polyubiquitinylation of DELLA and its degradation via the 26S proteasome pathway (see highlight lines 291-294). Then, downstream genes responding to GA would be released. Interestingly, PFD subunits could be interacted with DELLA by their subcellular localization shuttle, therefore to mediate DELLA protein homeostasis (details see the paragraph of 3.3.1). For auxin signaling, only one study showed that PFDs could be regulated by auxin treatment, and multiple ARF (auxin response factor)-binding sites, and auxin-inducible elements upstream in prefoldin genes locus. However, how PFDs mediate ARFs expression are still puzzling (see lines 317-326).
For gene expression manner, a very recent study revealed that this mechanism is much dependent on the action of PFDs in chromatin remodeling. Indeed, PFDs interact with components of SWR1c, mediating the exchange of H2A/H2B dimers with H2A.Z/H2B. This histone variants regulate nucleosome stability and chromatin structure, therefore downstream genes expression are subsequently modulated (see highlight line 229-230). However, from the structural biology perspective, how PFD-SWR1 complex mediate chromatin remodeling conformation remains to be elusive further.
For the main text of the re-submitted manuscript, please refer to the attachment.

Reviewer 2 Report
Comments and Suggestions for Authors
This review covers the prefoldin family, whose members have roles in protein folding and gene expression across different branches of life. It is well-organized, providing an engaging introduction for those new to the topic and a thorough overview of recent developments for experts. Prefoldins have been more extensively studied in yeast and animal cells as compared to plants, and this review highlights the emerging knowledge and outstanding questions for prefoldins in plants and across species. The review covers the structures and functions of prefoldin complexes, including both the protein folding and gene expression activities of the prefoldins, the latter particularly in relation to major plant master regulators auxin and gibberellin. This excellent review will be of great interest to the readership of Plants.
Author Response
Comment: This review covers the prefoldin family, whose members have roles in protein folding and gene expression across different branches of life. It is well-organized, providing an engaging introduction for those new to the topic and a thorough overview of recent developments for experts. Prefoldins have been more extensively studied in yeast and animal cells as compared to plants, and this review highlights the emerging knowledge and outstanding questions for prefoldins in plants and across species. The review covers the structures and functions of prefoldin complexes, including both the protein folding and gene expression activities of the prefoldins, the latter particularly in relation to major plant master regulators auxin and gibberellin. This excellent review will be of great interest to the readership of Plants.
Response to this Reviewer:
We sincerely thank the reviewer for careful reading. We appreciate the enthusiasm and the patience of this reviewer. Our manuscript has been polished by a native speaker to improve some writing and make it easier for readers to comprehend (see section of acknowledgment). We hope this review could provide a novel interest to the readership of plant biology-related fields.
For the main text of the re-submitted manuscript, please refer to the attachment.
